# A Critical Examination of Academic Hospital Practices—Paving the Way for Standardized Structured Reports in Neuroimaging

**DOI:** 10.3390/jcm13154334

**Published:** 2024-07-25

**Authors:** Ashwag Rafea Alruwaili, Abdullah Abu Jamea, Reema N. Alayed, Alhatoun Y. Alebrah, Reem Y. Alshowaiman, Loulwah A. Almugbel, Ataf G. Heikal, Ahad S. Alkhanbashi, Anwar A. Maflahi

**Affiliations:** 1Radiological Sciences Department, King Saud University, Riyadh 11451, Saudi Arabia; 2Scientists Unit, Central Research Laboratory, King Saud University, Riyadh 11495, Saudi Arabia; 3College of Medicine, King Saud University, Riyadh 11451, Saudi Arabia; 4Department of Radiology and Medical Imaging, King Saud University Medical City, Riyadh 145111, Saudi Arabia; 5Magnetic Resonance Imaging Unit, King Faisal Specialist Hospital & Research Center (KFSHRC), Riyadh 12713, Saudi Arabia; 6Cath Lab, Radiological Imaging, Prince Sultan Cardiac Center, Riyadh 11625, Saudi Arabia; 7Radiology Department, Ad Diriyah Hospital, Riyadh 13717, Saudi Arabia; 8Health Gates—Center of Excellence for Primary Health Care, Riyadh 12214, Saudi Arabia

**Keywords:** structured reporting, neuroimaging, radiologists, text report, MRI, narrative report

## Abstract

**Background/Objectives:** Imaging studies are often an integral part of patient evaluation and serve as the primary means of communication between radiologists and referring physicians. This study aimed to evaluate brain Magnetic Resonance Imaging (MRI) reports and to determine whether these reports follow a standardized or narrative format. **Methods:** A series of 466 anonymized MRI reports from an academic hospital were downloaded from the Picture Archiving and Communication System (PACS) in portable document format (pdf) for the period between August 2017 and March 2018. Two hundred brain MRI reports, written by four radiologists, were compared to a structured report template from the Radiology Society of North America (RSNA) and were included, whereas MR-modified techniques, such as MRI orbits and MR venography reports, were excluded (n = 266). All statistical analyses were conducted using Statistical Package for the Social Sciences (SPSS) statistical software (version 16.4.1, MedCalc Software). **Results:** None of the included studies used the RSNA template for structured reports (SRs). The highest number of brain-reported pathologies was for vascular disease (24%), while the lowest was for infections (3.5%) and motor dysfunction (5.5%). Radiologists specified the Technique (n = 170, 85%), Clinical Information (n = 187, 93.5%), and Impression (n = 197, 98.5%) in almost all reports. However, information in the Findings section was often missing. As hypothesized, radiologists with less experience showed a greater commitment to reporting additional elements than those with more experience. **Conclusions:** The SR template for medical imaging has been accessible online for over a decade. However, many hospitals and radiologists still use the free-text style for reporting. Our study was conducted in an academic hospital with a fellowship program, and we found that structured reporting had not yet been implemented. As the health system transitions towards teleservices and teleradiology, more efforts need to be put into advocating standardized reporting in medical imaging.

## 1. Introduction

Imaging studies are often an integral part of patient evaluation, and the associated radiology reports are part of patient management, which serves as the primary means of communication between radiologists and referring physicians. Therefore, completeness and effectiveness are critical for delivering accurate and easily understandable imaging results. In traditional reporting, radiologists customize the organization and content of a report for a specific case. The inherent variability between unstructured radiology reports may lead to certain reports being viewed as more effective than others owing to their unstructured nature. Structured reporting aims to standardize the format used in imaging reports [1]. This standardization may serve to increase report completeness and thus effectiveness. Several studies advocate the use of structured report (SR) templates to improve the quality of radiological reports [2,3]. Beyond improvements in the consistency of structure, language, and accuracy, promoting the standardized documentation of discrete data elements is necessary to optimize the value of the report and guide patient care management decisions [4]. The Michigan Radiology Quality Collaborative found that template reporting of brain MRI examinations increases the rate at which multiple sclerosis (MS)-relevant findings are included in the report and are preferred by MS neurologists [1]. Templates for SRs are freely available online and contain multiple features [5,6]. Nine key features have been found more frequently in SRs than in non-structured reports for MS [1]. With more features, reports convey sufficient information for making adequate clinical decisions more frequently than non-structured reports [7]. The literature indicates a preference for SRs among physicians. Consultants from multidisciplinary teams who regularly interact with SRs show greater satisfaction with the quality and content of these reports. However, support from radiologists was comparatively lower [8]. The most commonly cited barriers to SR adoption among radiologists include the length of SRs, limited awareness by many radiologists, and the absence of consensus regarding adoption within practices [9]. The advantages of SRs have been well recognized in the literature. Surgeons found that adding 13 key features to reports answered their questions in 98% of the SRs compared to 48% of the free-text (FT) reports. In addition, SRs show a significantly higher average of reported key features and less variability among reports than FT reports [10]. Gynecologists, interventional radiologists, and diagnostic radiologists found that structured MRI reporting templates of uterine fibroids with 19 key features had a much lower chance of missing essential information compared with FT reporting due to the restricted format and consistent nomenclature [11]. These reports were described as more helpful for surgical planning and easier to understand than narrative reports. MRI SRs for patients with rectal cancer facilitate surgical planning and lead to a higher satisfaction level of referring surgeons with more confidence in correctness compared to FT reports [10]. Similar findings were reported by surgeons using computed tomography (CT) SRs [12]. By employing standardized terminology and report structures, SRs maintain consistency across various reports and radiologists, which is essential for comparing patient outcomes over time and across different facilities.

The effectiveness of structured reporting has been demonstrated using several radiological procedures [13]. In interventional radiology procedures, compliance with reporting fluoroscopy time, administration of contrast, and radiation dose is improved by structured reporting [14]. SRs provide thorough details in less time and are easier to complete than conventional reports [14]. For inexperienced physicians or radiologists or those in training, SRs offer a consistent and educational format [15] that helps them understand what information is crucial and how to document clinical findings systematically. The uniformity in data presentation within SRs facilitates data extraction and analysis. This is particularly useful for research and in feeding clinical databases that support large-scale studies for evidence-based medical practices. Reviewing the literature leads to wide agreements on the effectiveness and accuracy of SRs compared to FT reports in clinical decision-making, with more helpful and less missing information and easier understanding. Systematic standards and criteria should be used when writing SRs. The current study aimed to assess the quality of free-form brain MRI reports with reports on standard reporting templates.

## 2. Materials and Methods

### 2.1. Sample of Reports

A series of 466 anonymized MRI reports from an academic hospital were retrospectively downloaded from the Picture Archiving and Communication System (PACS) in portable document format (pdf) for the period between August 2017 and March 2018 and served as the test set for this investigation. This time frame was specifically chosen to ensure that we collected the most recent reports available during the time of this study, thereby guaranteeing that the style and methodology of reporting were up to date. All reports during this time frame were downloaded and only those reports that met our predefined inclusion criteria for further analysis were set for this study. Inclusion criteria: only brain MRI reports, routine MR angiography of the brain, and MR perfusion (n = 200), whereas other MRIs such as shoulder, knee, abdomen, or any brain MR-modified techniques, such as MRI orbits and MR venography reports, were excluded (n = 266). The reports were created using voice dictation and transcribed using a speech recognition system. The report text represented the final report content approved by the MR radiologists and consisted of the procedure name, narrative findings, and impression. The focus was exclusively on brain MRI reports to maintain a clear and consistent scope. This focused approach enhanced the depth and precision of the analysis by limiting confounding factors, such as variability that could arise from adhering to structured report templates used in other types of MRI examinations. This strategy ensured that the findings were specific to brain MRI reporting styles, thereby providing a more controlled and accurate assessment of the data. Before carrying out the study, the protocol was approved by the Institutional Review Board of King Saud University-Medical City (KSU-MC), Riyadh, Saudi Arabia.

### 2.2. Sample of Radiologists

The included reports were published by four radiologists: two radiologists (RAD 1 and RAD 3) with more than 20 years of experience each and the others (RAD 2 and RAD 4) with less than 15 years of experience.

### 2.3. Standard Template

The RSNA Radiology Reporting Template for brain MRI was used as the reference standard for structured reporting (Figure 1) [6]. The MRI brain template contains 22 key features that must be reported to guide radiologists in formulating reports in best practice. We extracted SR elements from each report into an Excel file. The template includes five primary sections, each with sub-elements. The “MR Brain” section contains three sub-elements, incorporating the reported technique’s name and the use of IV contrast. The “Findings” section, the fourth section, contains 16 sub-elements. The presence or absence of each element was determined based on its mention in the narrative report, irrespective of the finding being normal or abnormal. If a particular element was not mentioned in the narrative report, it was considered absent. Data were extracted from the reports and divided into columns according to the template elements.

### 2.4. Statistical Analysis

Descriptive statistics were generated for the demographic and clinical characteristics of the patients. Comparisons between each key feature were performed using the chi-squared or Fisher’s exact test in brain MRI reports. Data were presented as the frequencies for categorical variables. Statistical significance was set at *p* < 0.05. All statistical analyses were conducted using SPSS statistical software (version 16.4.1, MedCalc Software). Contingency tables were used to determine the relationship between the finding variables and radiologists’ years of experience.

## 3. Results

### 3.1. Population

The study included 200 brain MRI reports that were written and approved by four radiologists and downloaded from one academic hospital. We analyzed all narrative reports and compared their details with those of the RSNA SR template. No significant differences were observed between reports of male and female patients and patients who underwent multiple examinations that were included in the study, which explains why the total number of reports (n = 200) is greater than the total number of patients (n = 179) (Table 1).

Report characteristics are presented in Table 2, with different types of pathologies. Pathological cases represented most of the reports (74%), whereas 26% represented unremarkable cases. The highest number of brain-reported pathologies was for vascular disease (24%) and the lowest was for infections (3.5%) and motor dysfunction (5.5%). Other pathologies such as tumors (19%), structural abnormalities (12.5%), and neurodegenerative disease (9.5%) were reported in the included sample but were less frequent than vascular disease.

For the Technique section, we categorized reports based on imaging protocols applied for examination into three protocols; the most common were for routine MRI brain protocol (90%) and MRI angiography protocol (9.5%), while the least was for MRI perfusion protocol (0.5%) (Table 2).

### 3.2. Key Features

For brain MRI, none of the included reports (n = 200) followed the SR template. The Technique (n = 170, 85%), Clinical Information (n = 187, 93.5%), and Impression (n = 197, 98.5%) were recorded in almost all reports. The Comparison section was mentioned in half of the reports. The Findings section had more missing information than the other sections. In particular, elements such as axial space (n = 191, 95.5%), basal cisterns (n = 183, 91.5%), cerebellum (n = 142, 71%), brain stem (n = 154, 77%), calvarium (n = 195, 97.5%), visualized upper cervical spine (n = 139, 69.5%), Sella (n = 193, 96.5%), skull base (n = 199, 99.5%), and marrow (n = 198, 99%) were not mentioned and lacking the most. Some sub-elements, such as the paranasal sinus and mastoid air cells (n = 84, 42%) and visualized orbits (n = 106, 53%), were mentioned in less than half of the reports. Parenchyma is a very important part of any brain report; however, it was mentioned in only 119 (59.5%) reports (Table 3).

### 3.3. Years of Experience

Radiologists with more than 20 years of experience wrote 80 reports; radiologists 1 and 3 wrote 26 and 54 reports, respectively (Table 1). The remaining reports were written by other radiologists (n = 120) with less than 15 years of experience. Some included radiologists within academic hospitals who frequently undertake dual responsibilities, engaging in educational, research, and clinical roles. Academic commitments, including teaching, research activities, and administrative tasks, may redirect time and focus away from clinical practice, possibly resulting in an unequal distribution of radiological reports between the included radiologists.

There was a relationship between years of experience and some report elements, where radiologists with the least experience showed higher commitment to report extra elements such as techniques (*p* < 0.001) and more information in the Findings compared with other sections, including extra-axial space (*p* < 0.05), intracranial hemorrhage (*p* < 0.001), basal cisterns (*p* < 0.006), cerebellum (*p* < 0.001), brain stem (*p* < 0.05), paranasal sinus and mastoid air cells (*p* < 0.001), visualized orbits (*p* < 0.001), and visualized upper cervical spine (*p* < 0.001). Other elements showed no significant relationship between years of experience and fulfillment of all elements (Table 4).

### 3.4. Pathology

The relationship between reported adherence to the template and different pathologies was significant. We found that many sections and elements were mentioned when there was structural damage or motor dysfunction, such as tumors, infections, vascular disease, or neurodegenerative changes. Sections that were found to be significantly different based on reports of pathological studies are Intravenous Contrast (*p* < 0.0001), Comparison (*p* < 0.002), and Impression (*p* < 0.001) and Findings elements such as ventricular system (*p* < 0.011), parenchyma (*p* < 0.001), midline shift (*p* < 0.019), cerebellum (*p* < 0.035), brain stem (*p* < 0.004), vascular system (*p* < 0.001), paranasal sinus and mastoid air cells (*p* < 0.009), and visualized upper cervical spine (*p* < 0.001) (Table 5).

## 4. Discussion

This study investigated and compared the structure of brain MRI reports from an academic hospital with an SR template from the RSNA. We chose RSNA templates as they met the criteria for good SRs [13].

Our results showed wide variation in writing style, length, and content. The main findings confirm the hypothesis that narrative reports frequently miss many key elements that could aid in the management of neurological cases. None of the studied MRI reports followed this template, which is consistent with multiple previous results from other MR examinations [10,16]. Recognizing the significant limitations in the current state of FT reporting is crucial, as previous studies have favored SRs as helpful for treatment planning with less missing information and better understanding [11,17]. SRs not only eliminate further inquiries by clinicians or further examination requests but also reduce the level of incorrect diagnoses and treatment plans [18]. This is beneficial for interventional radiologists and referring physicians [1].

Of the total 200 reports, 169 (84.9%) included a technical description of the examination, and only 44 (22%) did not provide any information about intravenous contrast. This is in agreement with a previous study in which academic hospitals were found to include more extensive technical information than community hospitals [19]. Clinical information improves interpretation accuracy and reporting confidence [7]. Notably, this information was present in nearly all reports (184 (92.4%)). One would assume that this shows that the referring physicians provide clinical information/clinical history in the MRI request forms. Regarding the template, the Findings section had 16 sub-elements with midline shift in 146 (73.4%) and intracranial hemorrhage in 117 (59%) reports, which are common references and landmarks for most brain pathological findings [20] documented. However, other details related to anatomical findings were lacking or missing in most of the reports (52.8–99.5%). The information provided to physicians in this section is one of the most important elements for understanding imaging reports. Most physicians prefer detailed findings, such as whether the brain region appears normal and what the size of the abnormal structure is, to be documented in a tabulated format [21]. Surprisingly, the Impression section was fulfilled in all except four reports, of which one was a suboptimal study owing to physical motion. The impression section of a radiology report is crucial for delivering high-quality clinical care. It synthesizes imaging findings into a coherent interpretation that may suggest a diagnosis. This section should be clear and impactful and utilize language that is comprehensible, memorable, and actionable [22]). The Comparison section was mentioned only in half of the reports by radiologists, while the Findings section had more missing information than the other sections. These findings suggested that the length of the report is the main barrier to completing the report or the lack of awareness of radiologists [9]. Clinicians favor a structured style over a free-form impression for enhanced interpretation of results [3] and are in favor of itemized reports because they ensure comprehensive documentation of information in a radiology report [23]. Effective reporting in radiology not only presents factual observations clearly and concisely but also avoids unnecessary complexity and redundancy. Structured reporting, as we described, involves standardizing the structure, style, and lexicon of reports. This systematic approach aids in ensuring that reports are focused, timely, and concise, regardless of the individual radiologist’s level of experience.

During extensive training, radiologists systematically navigate through medical images to interpret the findings by observing and summarizing the pathology results. In many cases, radiologists are not trained to write good radiology reports in a structured training program. Instead, they adopt their trainer’s style, and, with experience, they develop and create their own FT reporting style. This “apprenticeship model” was introduced by Steele et al., assuming that more experienced radiologists’ reporting skills are better than those of their less experienced counterparts [24]. We investigated other factors that may contribute to reporting completeness, such as the radiologists’ experience and the type of pathology reported. Years of experience can contribute to the completeness of reports, but this common notion that experienced radiologists write better reports is contradicted by the finding that experienced radiologists write short FT reports [19,25]. These studies support our finding that reports approved by less experienced radiologists include more elements than those approved by more experienced radiologists. This may explain the resistance of experienced radiologists to SRs as a rigid framework [26]. These interfaces can help less experienced radiologists by guiding them through the procedure for accurate diagnosis and treatment. However, more experienced radiologists seem to find that these interfaces are a substantial hindrance to their productivity [24]. As their level of skill grows, radiologists learn to exclude a great deal of extraneous information from their analyses based on their comprehension of the composition and content of radiological images. As a result, experienced radiologists focus their attention and eye movements more accurately on pertinent sections of the image rather than examining every area equally [27]. While this increases efficiency, it can also obscure unanticipated results. A preference was shown for a structured list of short descriptions over “free text”, even in simpler examinations, such as ankle radiographs [28].

The nature of the pathology significantly contributes to the completeness of the report. This was evident in our study from reports of structural damage and motor dysfunction, such as tumors, infections, vascular disease, and neurodegenerative changes. This bias in writing narrative reports underscores the need for structured reporting and following disease-specific templates in report composition. Radiologists and physicians prefer SRs over FT reports because the former improves the transmission of high communication standards and, therefore, improves patient care [29]. The main cost-effectiveness advantage of the structured report is that additional imaging requests decreased by 43% [14]. These global endorsements of SRs have been repeatedly reported in all medical imaging modalities and are considered potential solutions for best-quality radiology reports [30]. The European Society of Radiology published a paper outlining the reasons for moving forward with structured reporting [31] and detailed know-how [32], which we recommend reading. Therefore, there is a need to adopt a structured approach to teaching and reporting. This style upholds the key to success by providing excellent information and ultimately improving the diagnostic value of the imaging reports.

To our knowledge, this is the first study to assess the structure of MRI reports in Saudi Arabia, and none of the reports followed a structured format. Subsections of the findings of these non-structured reports show that the probability of providing sufficient information to make adequate clinical decisions is low. Based on evidence, sufficient information for understanding and making adequate clinical decisions is high in SRs [1]. In addition, the number of missing findings could be avoided if structured reporting was followed, leading to increased quality of radiology reports, according to a previous study by Sahni et al. [4]. We believe that discussing the importance of structured reporting over FT reporting is overdue, as the international radiological community has established disease-specific structured reporting [33].

Although this study was conducted carefully, it has some limitations. The template used for the brain MRI report was the old version of the RSNA, which was available when this study was conducted. These new versions are disease-specific structured templates. A valid measure of the requesting physician’s satisfaction with these reports is necessary to make a clinical decision, which we are currently working on. A quality assessment tool for the obtained narrative reports is required, such as the tool by Lee et al. [25] or investigating the number of suggested further examinations that the patient underwent after MRI until a diagnosis was reached. This indicated whether the physicians were satisfied with the MRI reports and results.

In the future, the value of disease-specific template-based reporting in imaging examinations may be extended to different pathological and body MRI examinations. A comparison between reports from private and government hospitals can be beneficial because healthcare is free in Saudi Arabia. Investigating physician satisfaction from current FT reports is vital before advocating for SRs. It can also be used to evaluate the value of structured reporting systems for specific diseases.

## 5. Conclusions

We objectively evaluated the presence of key features from the structured report template in MRI brain narrative reports. We found that academic hospitals do not follow a systematic approach to writing MR brain reports. Narrative reporting has drawbacks that have led to global agreement on the necessity of moving towards a systematic and structured reporting style with disease-specific templates. However, subjective evaluation of original narrative MRI reports by clinicians and referring physicians is required.

## Figures and Tables

**Figure 1 jcm-13-04334-f001:**
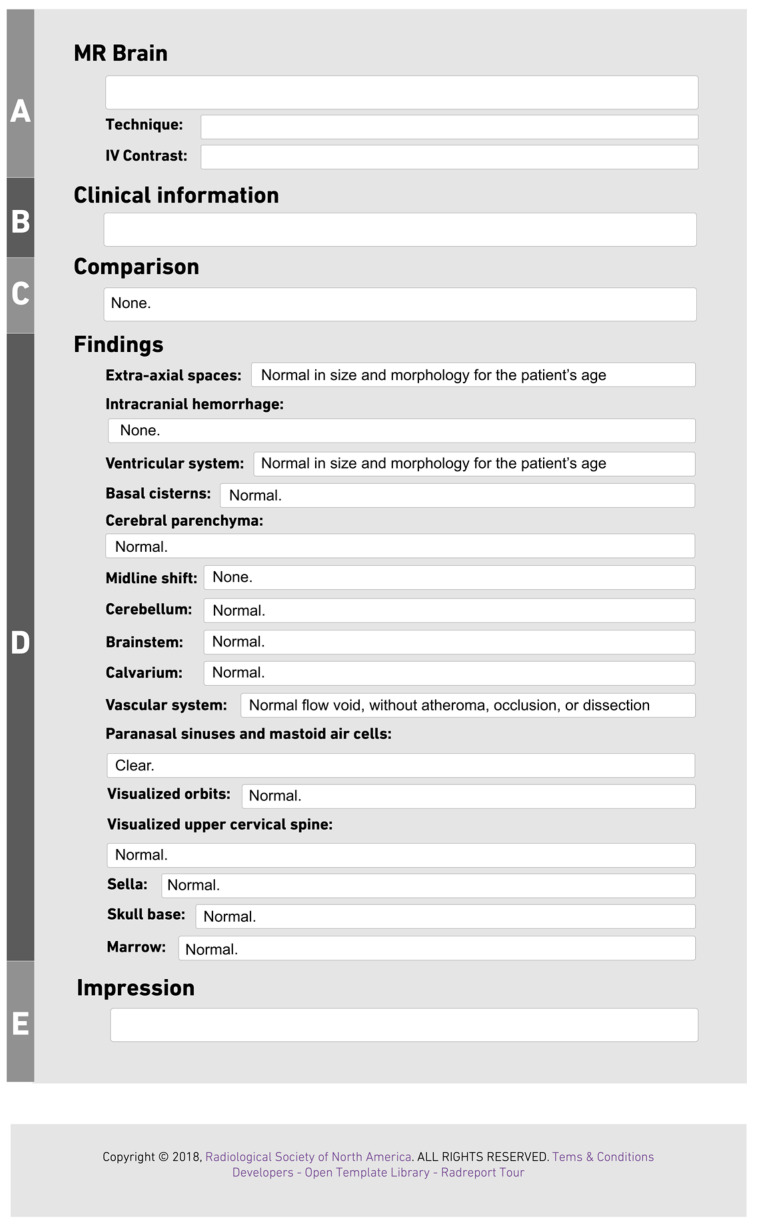
The template is divided into five sections: (**A**) MR Brain, where the technique and contrast are to be reported, (**B**) Clinical Information, (**C**) Comparison, (**D**) Findings, which is subdivided into 16 key features, and (**E**) Impression.

**Table 1 jcm-13-04334-t001:** Population demographics.

Sample	N
Brain MRI reports	200
Patients (Male:Female)	178 (86:92)
Patients’ age (mean ± SD)	39 ± 23
Reports by radiologists with experience of less than 15 years	120
Reports by radiologists with experience of more than 20 years	80

**Table 2 jcm-13-04334-t002:** Included report characteristics.

	Parameter	Frequency (n = 200)	Percentage (%)
Pathology	Normal	52	26
	Tumor	38	19
	Motor dysfunction	11	5.5
	Infections	7	3.5
	Vascular disorder	48	24
	Degenerative disorder	19	9.5
	Structural abnormality	25	12.5
Protocol	Routine MRI brain protocol	180	90
	MR angiography protocol	19	9.5
	MRI perfusion protocol	1	0.5

**Table 3 jcm-13-04334-t003:** Structured report elements followed by reports.

Elements	Sub-Elements	Mentioned n (%)	Not Mentioned n (%)
Technique		170 (85)	30 (15)
IV Contrast		116 (58)	116 (58)
Clinical Information		187 (93.5)	13 (6.5)
Comparison		100 (50)	100 (50)
Findings	Extra-axial space	9 (4.5)	191 (95.5)
	Intracranial hemorrhage	120 (60)	80 (40)
	Ventricular system	138 (69)	62 (31)
	Basal cisterns	17 (8.5)	183 (91.5)
	Parenchyma	119 (59.5)	81 (40.5)
	Midline shift	149 (74.5)	51 (25.5)
	Cerebellum	58 (29)	142 (71)
	Brainstem	46 (23)	154 (77)
	Calvarium	5 (2.5)	195 (97.5)
	Vascular system	61 (30.5)	139 (69.5)
	Paranasal sinus and mastoid air cells	84 (42)	116 (58)
	Visualized orbits	94 (47)	106 (53)
	Visualized upper cervical spine	67 (33.5)	133 (66.5)
	Sella	7 (3.5)	193 (96.5)
	Skull base	1 (0.5)	199 (99.5)
	Marrow	2 (1)	198 (99)
Impression		197 (98.5)	3 (1.5)

**Table 4 jcm-13-04334-t004:** The relationship between radiologist’s years of experience and the elements mentioned.

Element		Less than 15 Years N (%)		More than 20 Years N (%)		*p*-Value
		Mentioned	Not Mentioned	Mentioned	Not Mentioned	
Technique		115 (95.8)	5 (4.2)	55 (68.8)	25 (31.3)	0.000
IV Contrast		56 (46.7)	64 (53.3)	28 (35)	52 (65)	0.101
Clinical Information		113 (94.2)	7 (5.8)	74 (92.5)	6 (7.5)	0.640
Comparison		58 (48.3)	62 (51.7)	42 (52.5)	38 (47.5)	0.564
Findings	Extra-axial space	3 (2.5)	117 (97.5)	6 (7.5)	74 (92.5)	0.047
	Intracranial hemorrhage	89 (74.2)	31 (25.8)	31 (38.8)	49 (61.3)	0.000
	Ventricular system	81 (67.5)	39 (32.5)	57 (71.3)	23 (28.8)	0.831
	Basal cisterns	4 (3.3)	116 (96.7)	13 (16.3)	67 (83.8)	0.006
	Parenchyma	72 (60)	48 (40.0)	47 (58.8)	33 (41.3)	0.466
	Midline shift	94 (78.4)	26 (21.7)	55 (68.8)	25 (31.3)	0.099
	Cerebellum	24 (20)	96 (80.0)	34 (42.5)	46 (57.5)	0.000
	Brainstem	19 (15.9)	101 (84.2)	27 (33.8)	53 (66.3)	0.005
	Calvarium	1 (0.8)	119 (99.2)	4 (5.1)	76 (95.0)	0.163
	Vascular system	38 (31.7)	82 (68.3)	23 (28.8)	57 (71.3)	0.574
	Paranasal sinus and mastoid air cells	72 (60)	48 (40.0)	12 (15.1)	68 (85.0)	0.000
	Visualized orbits	79 (65.9)	41 (34.2)	15 (18.8)	65 (81.3)	0.000
	Visualized upper cervical spine	54 (45.8)	65 (54.2)	12 (15.0)	68 (85.0)	0.000
	Sella	3 (2.5)	117 (97.5)	4 (5.0)	76 (95.0)	0.290
	Skull base	1 (0.8)	119 (99.2)	0 (0.0)	80 (100)	0.413
	Marrow	2 (1.7)	118 (98.3)	0 (0.0)	80 (100)	0.246
Impression		117 (97.5)	3 (2.5)	80 (100.1)	0 (0.0)	0.084

**Table 5 jcm-13-04334-t005:** The relationship between the type of pathology and the mentioned elements.

Elements		Normal N (%)		Tumor N (%)		Functional Disorder N (%)		Infections N (%)		Vascular Disorder N (%)		Degenerative Disorder N (%)		Structural Disorder N (%)		*p*-Value
		M	NM	M	NM	M	NM	M	NM	M	NM	M	NM	M	NM	
Technique		43 (82.7)	9 (17.3)	31 (81.6)	7 (18.4)	8 (72.7)	3 (27.3)	7 (100)	0 (0.0)	44 (91.7)	4 (8.3)	16 (84.2)	3 (15.8)	21 (84)	4 (16)	0.569
IV Contrast		24 (46.2)	28 (53.8)	22 (57.9)	16 (42.1)	0 (0.0)	11 (100)	6 (85.7)	1 (14.3)	16 (33.3)	32 (66.7)	6 (31.6)	13 (68.4)	10 (40)	15 (60)	0.003
Clinical Information		45 (86.5)	7 (13.5)	35 (92.1)	3 (7.9)	11 (100)	0 (0.0)	7 (100)	0 (0.0)	47 (97.9)	1 (2.1)	18 (94.7)	1 (5.3)	24 (96.0)	1 (4)	0.288
Comparison		52 (100)	0 (0.0)	38 (100)	0 (0.0)	11 (100)	0 (0.0)	7 (100)	0 (0.0)	48 (100)	0 (0.0)	19 (100)	0 (0.0)	25 (100)	0 (0.0)	0.002
Findings	Extra-axial space	1 (1.9)	51 (98.1)	4 (10.5)	34 (89.5)	0 (0.0)	11 (100)	1 (14.3)	6 (85.7)	3 (6.3)	45 (93.8)	0 (0.0)	19 (100)	0 (0.0)	25 (100)	0.073
	Intracranial hemorrhage	32 (61.5)	20 (38.5)	16 (42.1)	22 (57.9)	6 (54.5)	5 (45.5)	5 (71.4)	2 (28.6)	31 (64.6)	17 (35.4)	13 (68.4)	6 (31.6)	17 (68)	8 (32)	0.556
	Ventricular system	39 (75)	13 (25)	24 (63.2)	14 (36.8)	7 (63.6)	4 (36.4)	4 (57.1)	3 (42.9)	37 (77.1)	11 (22.9)	13 (68.4)	6 (31.6)	14 (56)	11 (44)	0.011
	Basal cisterns	5 (9.6)	47 (90.4)	5 (13.2)	33 (86.8)	0 (0.0)	11 (100)	0 (0.0)	7 (100)	2 (4.2)	46 (95.8)	0 (0.0)	19 (100)	5 (20)	20 (80)	0.102
	Parenchyma	47 (90.4)	5 (9.6)	17 (44.7)	21 (55.3)	6 (54.5)	5 (45.5)	6 (85.7)	1 (14.3)	16 (33.3)	32 (66.7)	10 (52.6)	9 (47.4)	17 (77)	8 (32)	0.000
	Midline shift	45 (86.5)	7 (13.5)	20 (52.7)	18 (47.4)	10 (90.9)	1 (9.1)	5 (71.4)	2 (28.6)	39 (81.3)	9 (18.8)	13 (68.5)	6 (31.6)	17 (68)	8 (32)	0.019
	Cerebellum	15 (28.8)	37 (71.2)	10 (28.9)	27 (71.1)	2 (18.2)	9 (81.8)	0 (0.0)	7 (100)	11 (23)	37 (77.1)	10 (52.6)	9 (47.4)	9 (36)	16 (64)	0.035
	Brainstem	13 (25)	39 (75)	5 (13.2)	33 (86.8)	2 (18.2)	9 (81.8)	2 (28.6)	5 (71.4)	8 (16.7)	40 (83.3)	9 (47.4)	10 (52.6)	7 (28)	18 (72)	0.004
	Calvarium	1 (1.9)	51 (98.1)	1 (2.6)	37 (97.4)	2 (18.2)	9 (81.8)	0 (0.0)	7 (100)	1 (2.1)	47 (97.9)	0 (0.0)	19 (100)	0 (0.0)	25 (100)	0.073
	Vascular system	16 (30.7)	36 (69.2)	6 (15.8)	32 (84.2)	1 (9.1)	10 (90.9)	0 (0.0)	7 (100)	28 (58.3)	20 (41.7)	5 (26.3)	14 (73.7)	5 (20)	20 (80)	0.000
	Paranasal sinus and mastoid air cells	27 (51.9)	25 (48.1)	8 (21.1)	30 (78.9)	3 (27.3)	8 (72.7)	3 (42.9)	4 (57.1)	25 (52.1)	23 (47.9)	5 (26.3)	14 (73.7)	13 (52)	12 (48)	0.009
	Visualized orbits	24 (46.2)	28 (53.8)	12 (31.6)	26 (68.4)	4 (36.4)	7 (63.6)	3 (42.9)	4 (57.1)	28 (58.3)	20 (41.7)	10 (52.6)	9 (47.4)	13 (52)	12 (48)	0.113
	Visualized upper cervical spine	22 (42.3)	30 (57.7)	6 (15.8)	32 (84.2)	4 (36.4)	7 (63.6)	3 (42.9)	4 (57.1)	19 (39.6)	29 (60.4)	4 (21.1)	15 (78.9)	9 (36)	16 (64)	0.008
	Sella	1 (1.9)	51 (98.1)	4 (10.5)	34 (89.5)	1 (9.1)	10 (90.9)	0 (0.0)	7 (100)	1 (2.1)	47 (97.9)	0 (0.0)	19 (100)	0 (0.0)	25 (100)	0.543
	Skull base	0 (0.0)	52 (100)	1 (2.6)	37 (97.4)	0 (0.0)	11 (100)	0 (0.0)	7 (100)	0 (0.0)	48 (100)	0 (0.0)	19 (100)	0 (0.0)	25 (100)	0.638
	Marrow	1 (1.9)	51 (98.1)	1 (2.6)	37 (97.4)	0 (0.0)	11 (100)	0 (0.0)	7 (100)	0 (0.0)	48 (100)	0 (0.0)	19 (100)	0 (0.0)	25 (100)	0.859
Impression		52 (100)	0 (0.0)	36 (94.7)	2 (5.3)	11 (100)	0 (0.0)	6 (85.7)	1 (14.3)	48 (100)	0 (0.0)	19 (100)	0 (0.0)	25 (100)	0 (0.0)	0.000

M—Mentioned; NM—Not Mentioned.

## Data Availability

If required, data can be obtained from the corresponding author.

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
