# Peer review of "A Critical Examination of Academic Hospital Practices—Paving the Way for Standardized Structured Reports in Neuroimaging"

_jcm, 2024, doi:10.3390/jcm13154334_

Round 1

Reviewer 1 Report

Comments and Suggestions for Authors

Titel

A critical examination of academic hospital practices - paving 2 the way for standardized structured reports in neuroimaging.

Thanks for the opportunity for reviewing this interesting paper.

Introduction can benefit of a broader description of the value of structured reporting and perhaps why radiologist has barriers (e.g. Investigated in this study by Andersen et al, using structured templates or free text style in reporting CT staging on colon cancer: a national survey) or this can also be discussed in the discussion part. But it is important to discuss why – and if it should be mandatory using templates.

The aim is ok and relevant.

Methods

Please explain how the random selection of MRI reports was performed.

Who was responsible for the random selection?

Please include inclusion and exclusion criteria.

Why is MR brain relevant?

Results:

Any use of Contrast agents in MRI scans (Table 2), and Tesla type of MRI?

Was there errors between the 4 radiologists?

Author Response

Respected Editor,

We express our gratitude to all the reviewers for their comments. The manuscript has been revised based on the feedback provided by both reviewers. In conclusion, I would like to express my gratitude for the insightful points and suggestions provided by the reviewers. Their feedback has been instrumental in enhancing the clarity, depth, and overall quality of this manuscript. I have made every effort to address each comment thoroughly and have incorporated changes that significantly improve the research presentation and arguments. I believe these modifications have strengthened the paper and hope that it now meets the journal's standards for publication. Thank you once again for the valuable input and for the opportunity to refine this work.

Below is our response to the comments from Reviewer 1:

Comments that Reviewer 1 mentioned

New version Lines

Responses to Reviewer 1

1. Introduction can benefit of a broader description of the value of structured reporting and perhaps why radiologist has barriers (e.g. Investigated in this study by Andersen et al, using structured templates or free text style in reporting CT staging on colon cancer: a national survey) or this can also be discussed in the discussion part. But it is important to discuss why – and if it should be mandatory using templates

Lines 62-68

Lines 79-82

Lines 87-92

Thank you for your efforts and dedication in helping us enhance the quality of our manuscript. As recommended, we have included multiple paragraphs detailing the advantages of structured reporting over free text reports, as well as the challenges faced by radiologists based on evidence. These additions have been highlighted in Yellow for your convenience. Lines:

62-68

79-82

87-92

2.Please explain how the random selection of MRI reports was performed.

Lines 100-102

lines 103-106

The selection of MRI reports was not random; reports were retrospectively downloaded, after which non-brain MRI examinations (such as MRI orbits and brain venography) were excluded. Changes have been highlighted in yellow for your convenience. Edited as per below:

Lines 100-102

And lines 103-106

Inclusion criteria: only brain MRI reports, routine MR angiography of the brain, and MR perfusion (n=200), whereas other MRIs such as shoulder, knee, abdomen, or any brain MR-modified techniques, such as MRI orbits and MR venography reports, were excluded (n=266).

3.Who was responsible for the random selection?

Lines 103-107

In response to this query regarding the collection and selection of MRI reports, we would like to clarify that all MRI reports were downloaded from the PACS system during the period from August 2017 to March 2018. This time frame was specifically chosen to ensure that we collected the most recent reports available during the time of this study, thereby guaranteeing that the style and methodology of reporting were up-to-date. Following this initial collection, we then meticulously selected only those reports that met our predefined inclusion criteria for further analysis. This methodology ensured that our dataset was both comprehensive and representative of current reporting practices.

We edited the methods section with highlights in Yellow and add the following : (Lines 103-107)
This time frame was specifically chosen to ensure that we collected the most recent reports available during the time of this study, thereby guaranteeing that the style and methodology of reporting were up-to-date. All reports during this time frame were downloaded and only those reports that met our predefined inclusion criteria for further analysis were set for this study.

4.Please include inclusion and exclusion criteria.

Lines 107-110

Edited in the main manuscript as below: (lines 107-110)

Inclusion criteria: only brain MRI reports, routine MR angiography of the brain, and MR perfusion (n=200), whereas other MRIs such as shoulder, knee, abdomen, or any brain MR-modified techniques, such as MRI orbits and MR venography reports, were excluded (n=266).

Required changes had been highlighted in Yellow

5.Why is MR brain relevant?

Lines 113-119

In response to your inquiry regarding our exclusive focus on brain MRI reports, this decision was driven by the specific objectives of our study, which aimed to explore and analyze patterns of writing MRI reports. Additionally, brain MRI is the most requested examination in our dataset, providing a substantial volume of data for rigorous analysis. Each body region has its own structured reports and to limit confounding factors such as variability from following other structured report templates used in different types of MRI examinations we decided to select only Brain MRI routing structured report template. This approach ensured that our analysis could more accurately assess the impact and effectiveness of reporting styles within the context of brain imaging alone.

We edited the methods section and added the following (Lines 113-119):
The focus was exclusively on brain MRI reports to maintain a clear and consistent scope. This focused approach enhanced the depth and precision of the analysis by limiting confounding factors, such as variability that could arise from adhering to structured report templates used in other types of MRI examinations. This strategy ensured that the findings were specific to brain MRI reporting styles, thereby providing a more controlled and accurate assessment of the data.

6.Any use of Contrast agents in MRI scans (Table 2), and Tesla type of MRI

Our study did include reports from some contrast-enhanced MRI examinations as detailed in Table 2. However, the magnetic field strength of the MRI machines was not considered a contributing factor in our analysis of reporting styles and therefore was not specifically mentioned. Our primary focus was on the content and structure of the reports rather than the technical specifications of the imaging equipment used.

Reviewer 2 Report

Comments and Suggestions for Authors

The manuscript “A critical examination of academic hospital practices paving the way for standardized structured reports in neuroimaging” reports an interesting study evaluating 200 anonymized brain MRI reports and determining the effective follow of standardized protocol by the reports through comparison with RSNA structured template. Surprisingly, this study reveals that none of the included studies used the RSNA template for structured reports (SR).

1.     Line 150-153; “In particular, elements such as axial space (n=191, 95.5%), basal cisterns (n=183, 91.5%), cerebellum (n=142, 71%), brain stem (n=154, 77%), calvarium (n=195, 97.5%), visualized upper cervical spine (n=139, 69.5%), Sella (n=193, 96.5%), skull base (n=199, 99.5%), and marrow (n=198, 99%) were lacking the most.” Authors are required to clarify how these % values represent lacking data or comparison is not mentioned.

2.     Table 4; “Section 3.3. Years of Experience” clearly reveals about radiologists. Table 4 should be omitted.

3.     Supplementary Table 1 can be included in main text.

4.     Authors are required to discuss this finding. “The Comparison section was mentioned in half of the reports. The Findings section had more missing information than the other sections.”

5.     Line 233-235, “These studies support our finding that reports approved by less experienced radiologists include more elements than those approved by more experienced radiologists.” This can be easily understood because experienced radiologist can focus on important findings and to the point report. Do authors elaborate about relation between the importance of finding and revealing every detail with the experience of the radiologists?

Minor corrections:

            Line 95; the protocol was approved was approved

            Line 126-127; patients under-went multiple examinations

Author Response

Respected Editor,

We express our gratitude to all the reviewers for their comments. The manuscript has been revised based on the feedback provided by both reviewers. In conclusion, I would like to express my gratitude for the insightful points and suggestions provided by the reviewers. Their feedback has been instrumental in enhancing the clarity, depth, and overall quality of this manuscript. I have made every effort to address each comment thoroughly and have incorporated changes that significantly improve the research presentation and arguments. I believe these modifications have strengthened the paper and hope that it now meets the journal's standards for publication. Thank you once again for the valuable input and for the opportunity to refine this work.

Please find below our response to the comments of Reviewer 2:

Comments that Reviewer 2 mentioned

New version Lines

Responses to Reviewer 2

1.In particular, elements such as axial space (n=191, 95.5%), basal cisterns (n=183, 91.5%), cerebellum (n=142, 71%), brain stem (n=154, 77%), calvarium (n=195, 97.5%), visualized upper cervical spine (n=139, 69.5%), Sella (n=193, 96.5%), skull base (n=199, 99.5%), and marrow (n=198, 99%) were lacking the most.” Authors are required to clarify how these % values represent lacking data or comparison is not mentioned.

Line 128

We appreciate your note as we missed to include Figure 1 during the submission which will make it easier to understand the results. In Figure 1, The MRI brain template contains 22 key features that must be reported to guide radiologists in formulating reports. The template includes five primary sections, each with sub-elements. The “MR Brain” section contains three sub-elements, incorporating the reported technique's name and the use of IV contrast. The "Findings" section, the fourth section, includes 16 sub-elements.

The percentages represent the non mentioned data of sub-elements in the reports. This is determined by counting the blank sub-elements (e.g. the axial space had not been mentioned in 191 reports by radiologists, out of 200 reports, which represents 95.5% of the total data, this calculation was applied to all sub-elements and the changes were highlighted

2. Table 4; “Section 3.3. Years of Experience” clearly reveals about radiologists. Table 4 should be omitted.

Line 190

Table 4 was removed. We edited Table 1 to include summary of reports reported by radiologists based on years of experience

3. Supplementary Table 1 can be included in main text.

Line 215-216

As suggested the supplementary table 1 is included in the main text as Table 5

4.Authors are required to discuss this finding. “The Comparison section was mentioned in half of the reports. The Findings section had more missing information than the other sections.

Lines 248-254

We have edited the discussion and added the following:

The Comparison section was mentioned only in half of the reports, by radiologists, while the findings section had more missing information than the other sections. These findings suggested that the length of the report is the main barrier to completing the report or the lack of awareness of radiologists (9). Clinicians favor a structured style over a free-form impression for enhanced interpretation of results, (3) as itemized reports ensure comprehensive documentation of information in a radiology report (22).

5.These studies support our finding that reports approved by less experienced radiologists include more elements than those approved by more experienced radiologists.” This can be easily understood because experienced radiologist can focus on important findings and to the point report. Do authors elaborate about relation between the importance of finding and revealing every detail with the experience of the radiologists?

Lines 259-262

Lines 276-286

we would like to clarify that our discussion emphasizes the implementation of structured reporting, not merely in terms of specific wording but as a systematic and consistent approach to crafting radiology reports.

We have edited the discussion and highlighted the edits in Yellow.

6.The protocol was approved was approved

Line 119

Thank you for the comment, we sincerely apologise for the error, as suggested it was corrected and the changes were highlighted in Yellow

7.Patients under-went multiple examinations

Lines 154-155

Thank you for the comment, we sincerely apologize for the error, as suggested it was corrected and the changes were Highlighted in Yellow

Kind regards,

Ashwag R. Alruwaili (on behalf of all authors)                   
